# Paving the way for rural revitalization: Empirical analysis of the 'Sihao Rural Road Policy' and transport mode choice in Inner Mongolia, China

Yintu Bao[1], Gumula Wuri[2], Dan Shan[2], Ligao Bao [3]*

1 Department of Economics, Inner Mongolia Open University, Hohhot, China, 2 United Graduate School of Agricultural Science, Tokyo University of Agriculture and Technology, Fuchu, Japan, 3 Faculty of Economics, Daito Bunka University, Itabashi, Japan

* bolagtuat@gmail.com

## Abstract

This study investigates the impact of the Sihao Rural Road (SHRR) Program on travel behavior in Qingshuihe County, Inner Mongolia, China focusing on travel mode choice, travel frequency, and travel distance. Using an integrated Structural Equation Model (SEM) and Discrete Choice Model (DCM) framework, data were collected from 127 households between August 2023 and March 2024. The analysis reveals that the SHRR program significantly reduces travel frequency, likely due to improved local accessibility that decreases the need for frequent trips. Simultaneously, SHRR facilitates longer travel distances and promotes greater reliance on motorized modes. This suggests that enhanced infrastructure enables residents to travel farther and more efficiently using private vehicles, motorcycles, and electric bicycles. Car ownership plays a critical role, significantly influencing both travel distance and the adoption of motorized modes. However, its relationship with the use of electric bicycles is more complex, with effects mediated by other factors such as travel distance and frequency. These findings underscore the importance of considering both direct and indirect effects of rural infrastructure policies when evaluating their impact on mobility patterns and transport mode choices.

## 1. Introduction

Over the past few decades, China has ascended to a dominant economic position, recording impressive growth rates that have transformed it into the world's second-largest economy. This meteoric rise, characterized by rapid industrialization and urbanization, has brought unprecedented prosperity to many regions within the country. However, accompanying this economic boom is the shadow of existing disparities and inequalities [1]. While coastal cities and major urban centers flourished, becoming global hubs of commerce and innovation, many rural areas lagged behind. This evident disparity between urban prosperity and rural stagnation highlighted the expanding divide in income and opportunities [2].

**Data availability statement:** All relevant data are within the manuscript and its Supporting Information files.

**Funding:** This work was supported by the Inner Mongolia Open University Special Project, "Research on the Impact of 'Sihao Rural Roads' Construction on Rural Economy in Inner Mongolia" [grant number IMOU-SSR2302]. The funders had no role in study design, data collection and analysis, decision to publish, or preparation of the manuscript.

**Competing interests:** The authors have declared that no competing interests exist.

In response to these challenges, the central government initiated a significant push in 2006, prioritizing rural road development as a means to foster a new socialist countryside [3]. The goal was clear: renovate village, improve rural infrastructure, enhance sanitation and housing, and bridge the urban-rural divide [4]. Central to achieving this objective was the enhancement of transportation systems, recognized as a pivotal catalyst for balanced development, ensuring that rural areas could actively engage in the broader economic story of the nation. Improved transportation infrastructure not only facilitates economic growth [5] but also significantly impacts the well-being of rural residents by providing better access to essential services such as healthcare [6], education [7], and markets [8]. These improvements lead to enhanced mobility and connectivity, which are crucial for the socio-economic integration of rural populations.

Among the myriad of infrastructure projects aimed at improving rural connectivity, the Sihao Rural Road Policy (SRRP) in China stand out as a significant intervention [9]. These projects are designed to enhance road quality and accessibility in rural areas, thereby facilitating better mobility and fostering economic growth. Understanding the impact of these projects is not only essential for evaluating the effectiveness of SRRP but also for informing future infrastructure policies in similar contexts.

The relationship between transportation infrastructure and travel behavior is a critical area of study, as it has profound implications for broader socio-economic well-being. Research demonstrates that well-planned infrastructure leads to increased accessibility and shifts in travel behavior, promoting diverse and sustainable modes of transport. For example, enhanced roads and public transit systems reduce travel time and enable more frequent trips, contributing to economic and social engagement [10]. Handy, Cao [11] finds that residents in neighborhoods with better transit accessibility tend to drive less. Strategic infrastructure planning, therefore, is not only about improving current transportation networks but also about anticipating and shaping future travel behaviors to support sustainable and resilient communities [12]. In addition, Allen and Farber [13] suggest that more coherent transportation planning could better capture social equity benefits, particularly when transit accessibility improves.

However, while the economic benefits of rural road infrastructure projects are well-documented, there remains a significant research gap in understanding how these projects influence the day-to-day transport mode choices of rural residents in China. Existing studies tend to aggregate the impacts of infrastructure development, thereby obscuring the behavioral changes that occur at the individual and household levels. This gap is particularly pronounced in the context of rural China, where comprehensive studies linking rural road improvements to changes in transport mode choice are sparse. Therefore, expanding the scope to look into rural road policy effects on transport mode choices in specific region is crucial.

The decision to focus on travel behavior stems from the critical role that daily mobility plays in realizing the broader benefits of rural road improvements. Enhanced rural roads can fundamentally reshape how often residents travel, how far they go, and which modes of transport they prefer. These changes in travel behavior mediate a wide range of socioeconomic outcomes: improved roads may reduce the time and

financial costs of commuting, prompt shifts in mode preference (e.g., from walking to motorized modes), and influence household decisions about vehicle ownership. By capturing these direct and indirect pathways, researchers gain a more comprehensive understanding of how the SRRP translates into real-world changes in mobility patterns—ultimately affecting economic opportunities, access to services, and social inclusion.

The SRRP, initiated in 2015, is particularly interesting because it aims to fix the differences in transport infrastructure between urban and rural areas [14]. By focusing on improving the quality and connectivity of rural roads, the policy aspires not only to make travel more convenient but also to foster new economic opportunities in rural areas. This could help close the persistent urban–rural income gap, a central concern for sustainable development. Through enhanced transport links, rural communities could see boosts in employment, market access, and overall well-being [15]. Despite the program's potential, there is still limited empirical evidence demonstrating how specific aspects of the SRRP—like upgraded road infrastructure and enhanced road management—translate into shifts in travel behavior and ultimately contribute to rural revitalization. Against this backdrop, our study provides a detailed empirical analysis of the impact of SRRP on transport mode choices among rural households in Inner Mongolia. Unlike previous studies that primarily focus on economic outcomes, we adopt a behavioral lens to examine how improvements in road infrastructure influence the preferences and decisions of rural residents regarding their transport modes, trip frequency, and travel distance. We utilize an integrated Structural Equation Modeling (SEM) and Discrete Choice Modeling (DCM) framework to capture both the direct and indirect effects of the SRRP on travel behavior, thus offering a more nuanced understanding of the mechanisms behind these changes.

In particular, the research addresses the following questions:

(1) Does the SRRP continue to exert significant direct effects on car ownership, travel frequency, travel distance, and travel mode choice after controlling for socio-demographic factors?

(2) How does the SRRP indirectly influence travel mode choice through its effects on mediating variables such as car ownership, travel frequency, and travel distance?

(3) What is the overall impact of the SRRP on travel mode choice, combining both direct and indirect effects?

By examining these questions, the study aims to provide a comprehensive perspective on how rural infrastructure improvements like the SRRP shape travel behavior. Findings from this research can inform policy recommendations for future rural road development and identify ways to optimize infrastructure investments to achieve broader socio-economic benefits and sustainable mobility patterns.

## 2. Sihao rural road policy and transport mode

### 2.1. Sihao rural road policy

In 2014, Chinese General Secretary Xi Jinping issued a significant directive emphasizing the need to better construct, manage, maintain, and operate rural roads. This directive was encapsulated in the "Sihao" goal, which translates to the four imperatives mentioned above. As a result of this targeted approach, regions across the country accelerated their rural road construction endeavors. Over the subsequent decade, approximately 2.53 million kilometers of new and renovated rural roads were established, offering a robust support structure to advance agricultural and rural modernization [16]. According to Ministry of Transport of China [17], By the end of 2022, the total length of rural roads reached 4.53 million kilometers, with 96% classified as graded roads. Within this timeframe, China backed the construction and renovation of 65,000 kilometers of resource roads, tourism routes, and industrial pathways in impoverished areas. All rural roads had essentially been included in the maintenance scope, with 89% achieving a rating of average to excellent quality. The guiding principle became clear: if there's a road, it must be maintained, and the maintenance must be thorough.

After 2014, several key policies were introduced, including the "Rural Road Construction Quality Management Measures", "Guidelines for Promoting High-Quality Development of 'Sihao Rural Roads'", and the "Medium and Long-term Development Outline for Rural Roads". The constant refinement of these policy systems facilitated the "Sihao Rural Road" construction to embark on a positive trajectory. By the end of 2022, China has a dedicated force of 668,000 individuals responsible for managing rural roads. This essentially translates to full coverage by the "road chief system" at the county administrative unit level tasked with rural road management. Cumulatively, 353 "Sihao Rural Road" national demonstration counties have been established, along with 49 integrated urban-rural transport demonstration counties. Additionally, 30 "Most Beautiful Rural Roads" have been named, continuously amplifying their leading and motivating role in the broader initiative.

### 2.2. Rural transportation systems and transport behavior

Travel behavior encompasses various aspects such as travel mode, frequency, distance, time, purpose, and the sequence of travel activities [18]. The main travel modes typically include cars, public transportation, and non-motorized options like walking or cycling [19]. Research has consistently demonstrated that different aspects of the built environment have varying impacts on travel behavior [20–22]. For instance, Handy, Boarnet [23] categorized the built environment into three components: land use patterns, urban design, and transportation systems. Therefore, to accurately understand travel behavior, it is essential to examine the specific characteristics of the built environment that directly influence individuals' travel mode choices.

In rural China, travel behavior studies have shown varying preferences depending on the region. For instance, urban–rural bus service quality has a significant positive impact on rural residents' bus travel-mode choice, more so than most other factors, except private car ownership in Henan Province [24]. In rural Sichuan, walking and electric bicycles are preferred, followed by private car and motorcycles [20]. In Jiangsu Province, social influence plays a significant role in shaping green travel behavior, neighbors who adopt green travel practices positively influence others, while those who do not engage in green travel have a negative impact [25]. Studies from within China underscore that the impact of transport infrastructure on travel behavior can vary considerably, influenced by the research areas selected and the unique facets of the specific infrastructure examined.

The travel patterns are further influenced by the age demographics, with younger people often commuting for work, while children and the elderly, with limited mobility, make fewer trips [26]. Research by Feng, Liu [27] using the MNL model found that travel distance significantly influences mode choice, whereas personal attributes and family income have less impact. Li, Zhang [28] explored that public transport has a direct negative impact on travel satisfaction in rural Sichuan. Studies by Ao, Chen [29], Ao, Yang [30] revealed that rural transportation systems significantly impact vehicle and private car ownership. Moreover, these systems also affect travel carbon emissions and influence the travel frequency of electric bicycles and motorcycles [31]. However, research has yet to establish a clear link between the SHRR program—focused on the construction, management, maintenance, and operation of rural roads—and travel mode choice in rural China.

## 3. Research method

### 3.1. Model structure

Structural Equation Modeling (SEM) is widely recognized as a powerful tool for analyzing travel behavior due to its ability to model complex relationships among variables [32]. Building on this, the integrated SEM and Discrete Choice Model (DCM) framework, as described by Ding, Wang [33], is particularly well-suited for examining the intricate interactions that influence travel mode choices. In this framework, key variables such as car ownership, travel frequency, and travel distance act as mediators that influence the relationship between the SHRR and the choice of travel mode, as shown in **Fig 1**. The model further incorporates socio-demographic characteristics and the impact of the SHRR project to

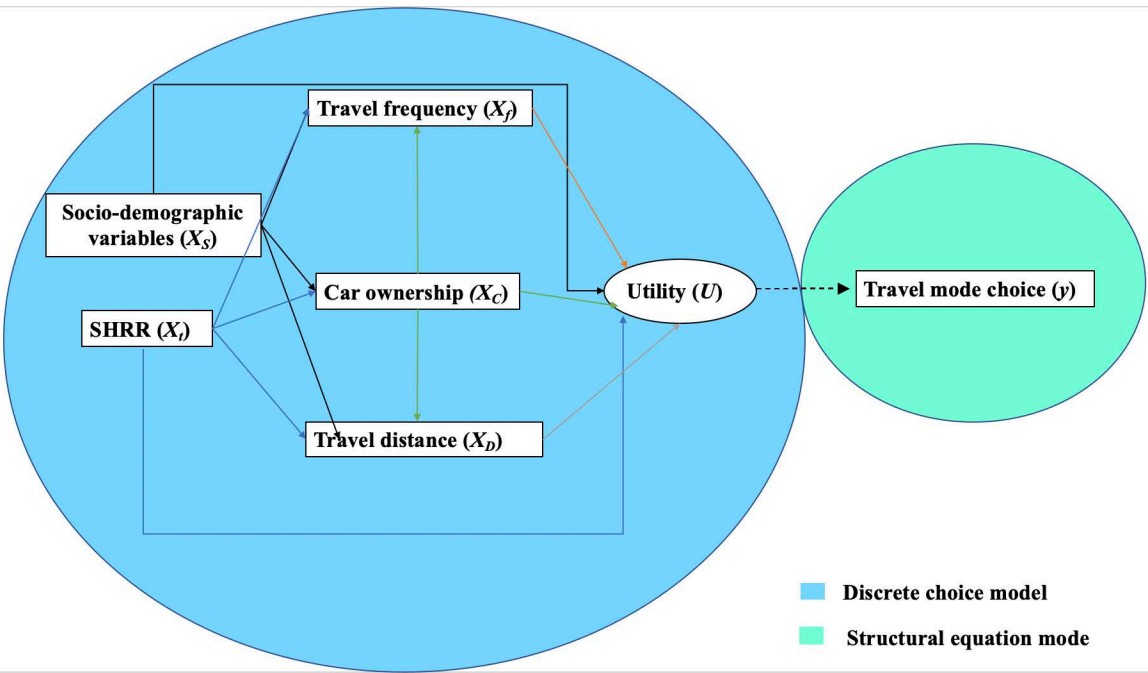

**Fig. 1. Research design for the integrated structural equation model and discrete choice model.**

better understand travel behavior. By including socio-demographic factors and individual attitudes, this approach helps to address the effects of residential self-selection, where personal preferences and characteristics shape both location choice and travel behavior [34].

The integrated model of SEM-DCM allows for the examination of both direct and indirect effects of the SHRR project on travel outcomes. The multinomial logit (MNL) model component is applied to evaluate choices among different travel alternatives, such as non-motorized modes (e.g., walking and cycling), electric bicycles, and motorized modes (e.g., cars, motorcycles). This integrated modeling structure provides a comprehensive understanding of how different factors collectively influence travel behavior and mode choice decisions.

### 3.2. Model specification

The integrated model consists of two main components: SEM component and DCM component. In this framework, endogenous variables are influenced by exogenous variables, either directly or through their interactions with other endogenous variables. The SEM component is particularly useful for describing and analyzing these complex relationships, capturing both direct and indirect effects within the system [35]. This integrated approach allows for a comprehensive examination of how various factors, such as socio-demographic characteristics and infrastructure improvements, collectively shape travel behavior.

$$\omega = B\omega + C\sigma + \epsilon \tag{1}$$

In the integrated model, the mathematical components in equation (1) are defined as follows: $\omega$ represents an $E \times 1$ matrix of endogenous variables, while $\sigma$ denotes an $X \times 1$ matrix of exogenous variables. The relationships between these variables are captured by two matrices: $B$, an $E \times E$ matrix that contains the coefficients for the endogenous variables, and $C$, an $X \times X$

matrix of coefficients for the exogenous variables. $\epsilon$ is the $E \times 1$ matrix representing the residuals or errors in the endogenous variables. It is important to note that in this study, all endogenous and exogenous variables are observed directly.

The Multinomial Logit (MNL) component of the model operates under the assumption that an individual $i$, when faced with a finite set $Si$ of alternatives $n$, selects the option $n$ that maximizes their utility, denoted by $U_{in}$. The utility for each alternative is modeled as a function of explanatory variables, which form the systematic part, combined with a disturbance term. The random utility function in this choice model can be expressed as a combination of these systematic components and random disturbances.

$$U_{in} = V(\omega, \sigma; \beta) + \varepsilon_{in} = \beta_\omega \omega_{in} + \beta_\sigma \sigma_{in} + \varepsilon_{in} \qquad (2)$$

In Equation 2, $\beta$ represents the parameters to be estimated for the observed exogenous and endogenous variables. The error terms, $\varepsilon$ are independently, identically distributed (i.i.d.) extreme value error terms. Based on the principle of maximum utility, the measurement equation for the observed choices can be formulated as follows:

$$y_{in} = \begin{cases} 1 \text{ if } U_{in} \geq U_{ip}, \ n \neq p \\ 0 \text{ otherwise} \end{cases} \qquad (3)$$

In Equation 3, $y_{in}$ is a choice indicator of whether alternative $n$ is chosen by individual $i$ or not. The likelihood function for an observation is then the joint probability of observing both the choice $y_{in}$ and the corresponding endogenous variables. The likelihood function for a given observation is the joint probability of observing the choice and the endogenous variables as follows:

$$P(y_n|\omega, \sigma, \beta_\omega, \beta_\sigma, C, \delta) = \int_{R_\omega} P_y(y_n|\omega, \sigma, \beta_\omega, \beta_\sigma, \varepsilon) f(\omega|\sigma, C, \epsilon) \, d\omega \qquad (4)$$

where $P_y$ represents the probability function of the choice model, and $f$ denotes the density function corresponding to the structural model for the endogenous variables. The term $\delta$ encompasses the full set of random errors within the model, while $R_\omega$ indicates that the integration is performed over the range space of the vector of endogenous variables. This formulation ensures that the likelihood function accounts for the joint distribution of both the observed choices and the endogenous variables, providing a robust framework for estimating the model parameters.

For estimating the parameters in both the SEM and DCM components, this study employed a simultaneous approach to integrate the models, following the methodology suggested by Raveau, Álvarez-Daziano [36].

### 3.3. The study area

Spanning approximately 1.2 million km², Inner Mongolia stands as China's third-largest administrative division, making up about 12% of the nation's total landmass. In 2019, its GDP per capita approached $10,000, positioning Inner Mongolia as the 11th wealthiest region in China [37]. As one of the provinces most devoted to the SRRP, Inner Mongolia earned the distinguished designation for 12 counties as National Demonstration Counties and propelled a further 38 counties to the status of Autonomous Region Level Demonstration Counties under the policy. By the end of 2022, rural and pastoral areas in Inner Mongolia had witnessed significant advancements in road infrastructure, with the total length of accessible roads reaching 175,000 kilometers [38]. This comprehensive network comprises 40,000 kilometers of county roads, 40,000 kilometers of township roads, and 95,000 kilometers of village roads. Every Sumu (an Inner Mongolian administrative unit equivalent to a township), township, eligible administrative village, and Gacha (a smaller Inner Mongolian administrative division below the level of a village) have been interconnected by hardened roads, with passenger vehicle services being made available.

Hohhot city, with a total area of 17,188.2 square kilometers, serves as a significant administrative and economic center in Inner Mongolia [39]. As of 2022, the city had a population of 3.55 million. The city is divided into four districts, four counties, one banner, and a national economic and technological development zone [40]. The city has a well-established transportation network, with a total road length of 8,635 kilometers, including highways and rural roads [41]. This network supports regional connectivity and accessibility, enhancing the socio-economic development of both urban and rural areas. Qingshuihe County, located within Hohhot City, covers an area of 2,817.5 square kilometers and is characterized by a well-developed transportation network, including national and provincial highways, as well as county and village roads [42]. By the end of 2022, the total road length in the county reached 1,840.876 kilometers, with a road density of 64.39 kilometers per 100 square kilometers [43].

The data for this study were collected in Qingshuihe County, focusing on rural residents both with and without access to the SHRR Program in two villages, during the period from August 2023 to March 2024. This study was conducted in accordance with the ethical standards of the institution's review board. Ethical approval for the research was granted by the Review Board of Inner Mongolia Open University [No. [2023] KY SSR2302], and all procedures were performed in compliance with relevant guidelines and regulations. We used a stratified sampling technique to select 200 households for the survey, which was distributed across four villages—village A, village B, village C and village D—in Qingshuihe County. However, due to incomplete questionnaires (n = 10), data inconsistencies (n = 18), and non-responses (n = 45), a total of 73 responses were excluded. As a result, the final sample for analysis comprised 127 valid responses. The collected data encompasses various aspects of travel behavior, including mode choices such as walking, bicycle, motorcycle, car, electric bicycle, public transport, tricycle, and tractor. Additionally, information on travel attributes, like average travel distance and frequency, was gathered. Demographic variables, including age, gender, income, and education, were also recorded to provide a comprehensive understanding of the factors influencing travel mode choices.

Table 1 presents key characteristics for the four study villages. Village A shows a moderate road density (1.15 km/km²), lower average income (14,500 yuan), and high reliance on agriculture (81%), suggesting a stronger dependence on traditional, local transport modes. Village B, which did not implement the SHRR, has a low road density (0.71 km/km²) but a high paved road percentage (93%) and an average income of 16,500 yuan—indicating that prior infrastructure investments have improved road quality, even if internal connectivity remains limited. In contrast, Village C stands out with the highest road density (2.3 km/km²), the lowest transport cost (5 yuan) and travel time (5 minutes) to the county center, an average income of 15,000 yuan, and a lower agricultural labor share (65%), reflecting superior connectivity and accessibility. Village D, which implemented the SHRR, shares a low road density (0.71 km/km²) with Village B but has the highest

**Table 1. Village characteristics.**

|  | Village A | Village B | Village C | Village D | Qingshuihe County |
|---|---|---|---|---|---|
| Village Area (km²) | 20 | 35 | 8.7 | 35 | 2818 |
| Road Density (km/km²) | 1.15 | 0.71 | 2.3 | 0.71 | 0.65 |
| Paved Road Percentage (%) | 85 | 93 | 90 | 85 | N/A |
| Average Transport Cost (Yuan) | 10 | 15 | 5 | 15 | N/A |
| SHRR | Yes | No | No | Yes | N/A |
| Road Upgrades in Last 5 Years (km) | 2 | 16 | 2 | 10 | 781 |
| Avg Travel Time to County Center (min) | 15 | 25 | 5 | 20 | N/A |
| Average income (Yuan) | 14500 | 16500 | 15000 | 17000 | 24110 |
| Population (People) | 434 | 998 | 1285 | 1742 | 136400 |
| Ratio of Labor Force Engaged in Agriculture (%) | 81 | 80 | 65 | 80 | N/A |

Source: Survey data and the Statistical Yearbook.

average income (17,000 yuan) and a moderate agricultural labor share (80%), along with higher transport costs (15 yuan) and longer travel time (20 minutes). For broader context, the average income in Qingshuihe County (24,110 yuan) is notably higher than that of the study villages, indicating economic disparities between rural and county-level populations. Moreover, the county's overall road density (0.65 km/km²) is lower than that of all study villages, as much of the county consists of non-residential land such as mountains, forests, and agricultural fields. This suggests that while rural areas face infrastructural challenges, localized road investments, including the SHRR program, may have enhanced connectivity in specific villages.

In the study area of Qingshuihe County, government-led initiatives such as targeted poverty alleviation, healthcare subsidies, and improvements to local schools and agricultural facilities have been implemented in recent years alongside the SHRR Program. To minimize potential confounding effects from these parallel policies, we employed two approaches. First, the sample villages were deliberately selected to include both households benefiting from SHRR improvements and those that did not, ensuring relative uniformity in exposure to other initiatives. Informal interviews with local officials and community leaders further confirmed that no major road-related or large-scale development programs were intro-duced during the study period that could disproportionately affect travel behavior. Second, the household survey included questions on recent policy-driven changes in income, agricultural activities, health services, education access, and vehicle ownership, enabling us to detect any unusual shifts unrelated to SHRR. These measures help ensure that our analysis more accurately isolates the impact of the SHRR program from other local influences.

### 3.4. Variable description

The travel mode choices of rural residents in Inner Mongolia include various options such as walking, bicycles, motorcy-cles, private cars, electric bicycles, tricycles, tractors, and public transportation. In this context, public transportation spe-cifically refers to the buses that connect villages, towns, and cities, providing essential links for rural residents to access broader economic and social opportunities. These modes reflect the diverse and context-specific needs of rural communi-ties in the region. The survey data show that rural residents are likely to choose motorcycle for their main daily travel, with a rate of up to 36.22%, followed by electric bicycle (19.69%), tricycle (17.32%), private car (11.81%), walking (10.24%), and bicycle (3.15%). Only 1.57% of rural residents chose tractor. The travel distance, travel frequency, and sociodemo-graphic variables are considered key factors that influence travel mode choice in a large number of studies [34]. Psycho-logical factors, such as preference, also affect the travel mode choice of residents [19].

Table 2 presents a comprehensive summary of the variables employed in the study to examine the impact of the Sihao Rural Road (SHRR) Policy on rural mobility in Inner Mongolia. The variables are categorized into several groups, including those related to travel behavior (e.g., *TM, Car_ownership, Average_travel_distance, Average_week_times_goout*), and sociodemographic factors (e.g., *Age, Education, Self_health, Incomelevel_22, Agri_work_share*). Additionally, the table includes variable that capture the influence of specific factors on travel habits over the past five years. These factors are reflected in the variable *Primary_factor_affecting_travel*, which identifies the attitudes that represents a main driver behind changes in respondents' travel modes and habits. The binary indicator *SHRR* provides insight into the infrastructural con-text of the households. This table serves as a foundational reference for understanding the relationships explored in the structural equation modeling and discrete choice modeling employed in the study.

Sociodemographic variables, including gender, age, highest education level of family head, and total household income, are considered in this study. Whether household own cars, motorcycles, and tractor is also considered as a reference. The survey data show that the number of male household heads is more than that of female head, female makes up only 3.15% of the respondents. Moreover, 75.59% of the respondents are over 50 years old because young and middle-aged residents are also likely to leave for work. For education level, 37.01% and 34.65% of the respondents have attained junior high school and primary school levels, respectively. Most of the sample households have two mem-bers who are working, with a total household income between 10000 and 50,000 yuan. Most of the sample households

**Table 2. Variable description.**

| Variable Name | Description |
| --- | --- |
| TM_3 | Main transport mode used by respondents in 2022, categorized by transport type: 1 = Non-motorized mode (walking and cycling), 2 = Electric bicycle, 3 = Motorized mode (car, motor, tractor). |
| Average_travel_distance | Average travel distance on a scale from 1 to 5, representing different distance ranges: 1 = Less than 1 km, 2 = 1–5 km, 3 = 6–10 km, 4 = 11–20 km, 5 = More than 20 km. |
| Average_week_times_goout | Average number of times respondents go out per week, represented as a continuous variable. |
| SHRR | Binary indicator of whether Sihao Rural Road (SHRR) projects were implemented in the village (1 = SHRR implemented before in village). |
| Age | Continuous variable representing the household head's age. |
| Incomelevel_22 | Household income level on a scale from 1 to 9, with higher values indicating higher income: 1 = Less than ¥10,000, 2 = ¥10,000-¥30,000, 3 = ¥30,000-¥50,000, 4 = ¥50,000-¥80,000, 5 = ¥80,000-¥120,000, 6 = ¥120,000-¥160,000, 7 = ¥160,000-¥200,000, 8 = More than ¥200,000, 9 = Prefer not to disclose or don't remember. |
| Agri_work_share | Household income share from agricultural work, indicating dependence on agriculture for livelihood. |
| Self_health | Self-reported health status of household head on a scale from 1 to 5, with higher values indicating better health: 1 = Very good, 2 = Good, 3 = Average, 4 = Poor, 5 = Very poor. |
| Education | Education level of the household head on a scale from 2 to 7, with higher values indicating higher education levels: 2 = Primary school, 3 = Middle school, 4 = High school, 5 = Vocational school, 6 = Junior college, 7 = University, 8 = Master's degree or higher. |
| Car_ownership | The indicator of household ownership of an electric bicycle, car, motor, or tractor (1 if owns any of these, 2 if owns more than one, sum based on presence of each vehicle). |
| Primary_factor_affecting_travel | The primary factor influencing the respondent's travel mode and habits over the past five years: 1 = Improvement in transportation facilities, 2 = Upgrade or purchase of vehicles, 3 = Relocation of home or workplace, 4 = Changes in economic conditions. |

have motorcycle, with electric bicycles accounting for the highest proportion, followed by bicycles, cars, and tractor. Table 3 provides additional detail regarding descriptive statistics.

## 4. Results and discussion

### 4.1. Descriptive analysis

The descriptive statistics in Table 3 provide a comprehensive overview of the variables in the study, highlighting key patterns in travel behavior and socio-demographic characteristics among rural residents in Qingshuihe County. The variable *TM_3* (Main Transport Mode) shows that 49.61% of respondents primarily use motorized modes (cars, motorcycles, tractors), 30.71% use non-motorized modes (walking, cycling, tricycles), and 19.69% rely on electric bicycles.s. The *Car_ownership* variable, with a mean of 0.969 and 65.35% of households own at least one vehicle (car, motor, tractor, or electric bicycle), while 15.75% own more than one. Only 18.90% of respondents reported not owning any vehicle.

The mean of *Average_week_times_goout* is 9.79, reflecting a range from 1 to 30 weekly outings and suggesting varied mobility patterns. *Average_travel_distance* has a mean of 2.803, indicating that most trips fall within the 6–10 km range. The majority of respondents (32.28%) travel between 11–20 km, followed by those who travel less than 1 km (25.98%) and 6–10 km (25.20%). Only 5.51% of respondents travel more than 20 km. The *SHRR* variable reveals that slightly more than half of the villages (mean = 0.559) have implemented SHRR projects, with a relatively balanced distribution across the sample.

The household heads' mean *Age* is 55.79 years, with most being middle-aged or older, and there is moderate variation in income levels (*Incomelevel_22* mean = 2.787), suggesting that the sample is primarily composed of low- to middle-income households. *Agri_work_share* indicates a significant reliance on agriculture, with a mean of 0.623, though this

**Table 3. Descriptive statistics.**

| Variables | Obs. | Mean/Percentage Share | S.D. | Min | Max |
|---|---|---|---|---|---|
| TM_3 | 127 | 2.188976 | 0.8795046 | 1 | 3 |
| Car_owners~p | 127 | 0.9685039 | 0.5900902 | 0 | 2 |
| Average_we~t | 127 | 9.787402 | 4.903238 | 1 | 30 |
| Average_tr~e | 127 | 2.80315 | 1.291236 | 1 | 5 |
| SHRR | 127 | 0.5590551 | 0.4984666 | 0 | 1 |
| Age | 127 | 55.7874 | 8.438978 | 30 | 76 |
| Incomelev~22 | 127 | 2.787402 | 1.051302 | 1 | 9 |
| Agri_work_~e | 127 | 0.6232283 | 0.2775771 | 0.1 | 1 |
| Self_health | 127 | 1.850394 | 0.7671711 | 1 | 5 |
| Education | 127 | 3.149606 | 1.241343 | 2 | 7 |
| Primary school | 44 | 34.65 | | | |
| Middle school | 47 | 37.01 | | | |
| High school | 24 | 18.9 | | | |
| Junior college | 9 | 7.09 | | | |
| University | 3 | 2.36 | | | |
| BK | 125 | 1.72 | 0.8576036 | 1 | 4 |
| Improvement in transportation facilities | | | | | |
| | 57 | 45.6 | | | |
| Upgrade or purchase of vehicles | | | | | |
| | 57 | 45.6 | | | |
| Changes in economic conditions | | | | | |
| | 11 | 8.8 | | | |

dependence varies across households. Meanwhile, the mean of *Self_health* is 1.850, suggesting generally good health among respondents. *Education* data reveal that 37.01% of household heads have completed middle school, and 34.65% have a primary school education. Finally, the *BK* variable (Primary Factor Affecting Travel) shows that 45.60% of household heads identified improvement in transportation facilities as the main factor, while another 45.60% cited the upgrade or purchase of vehicles. A smaller portion, 8.80%, indicated relocation of home or workplace as the primary factor.

### 4.2. Direct effects on car ownership

After controlling for individual' socio-demographics, the model results show that the SHRR still has a significant effect on household car ownership, as shown in Table 4. The variable *SHRR* has a significant direct negative effect on car ownership, with a coefficient of -0.2225 ($p < 0.05$). This indicates that households in areas where the SHRR program was implemented are less likely to own cars. This result suggests that improvements in rural road infrastructure reduce the need for car ownership, possibly by improving accessibility to alternative transportation modes, including public transit, shared mobility services, and non-motorized options. When road networks are upgraded, travel conditions for buses, ride-sharing services, and other public transport options improve, making them more efficient, reliable, and attractive alternatives to private car use. Additionally, better roads often attract investment in logistics, e-commerce delivery, and mobility-as-a-service (MaaS) solutions, providing further alternatives to car ownership. *Age* also shows a significant negative relationship with car ownership (coefficient = -0.0116, $p < 0.05$), implying that older individuals are less likely to own cars. This could reflect a decline in mobility needs or driving ability with age, or possibly greater dependence on alternative transport modes among older populations. *Incomelevel_22* has a significant negative effect on car ownership (coefficient = -0.1208, $p < 0.05$), contrary to the expectation that higher income might be associated with higher car ownership. This finding could suggest that

Table 4. Standardized direct, indirect and total effects on car ownership, travel frequency and travel distance.

| Variables | Car_ownership | Average_week_times_goout | | | Average_travel_distance | | |
|---|---|---|---|---|---|---|---|
| | Direct | Direct | Indirect | Total | Direct | Indirect | Total |
| | effect | effect | effect | effect | effect | effect | effect |
| Car_ownership | | −0.9083 | | −0.9083 | 0.5492** | | 0.5492** |
| | | (−1.25) | | (−1.25) | (−3.03) | | (−3.03) |
| Age | −0.0116* | −0.0057 | 0.0106 | 0.0048 | 0.0063 | −0.0064 | −0.0001 |
| | (−1.87) | (−0.11) | (−1.04) | (−0.09) | (−0.49) | (−1.59) | (−0.01) |
| Education | −0.034 | 0.5544 | 0.0309 | 0.5852 | −0.1002 | −0.0187 | −0.1189 |
| | (−0.69) | (−1.37) | (−0.6) | (−1.44) | (−0.99) | (−0.67) | (−1.14) |
| Self_health | −0.1179 | 0.2649 | 0.1071 | 0.3719 | 0.0213 | −0.0647 | −0.0434 |
| | (−1.54) | (−0.42) | (−0.97) | (−0.59) | (−0.14) | (−1.37) | (−0.27) |
| Agri_work_share | −0.0302 | −0.3697 | 0.0275 | −0.3422 | 0.2779 | −0.0164 | 0.2615 |
| | (−0.14) | (−0.21) | (−0.14) | (−0.20) | (−0.64) | (−0.14) | (−0.58) |
| Incomelevel_22 | −0.1208* | 0.1737 | 0.1097 | 0.2834 | 0.0126 | −0.0663 | −0.0537 |
| | (−2.36) | (−0.41) | (−1.11) | (−0.67) | (−0.12) | (−1.86) | (−0.50) |
| BK | −0.0487 | −0.1922 | 0.0443 | −0.148 | 0.1151 | −0.0268 | 0.0883 |
| | (−0.79) | (−0.38) | (−0.67) | (−0.29) | (−0.91) | (−0.76) | (−0.67) |
| SHRR | −0.2225* | −2.8012** | 0.2021 | −2.5991** | 0.9519*** | −0.1222 | 0.8297** |
| | (−1.78) | (−2.71) | (−1.02) | (−2.53) | (−3.69) | (−1.54) | (−3.15) |
| $R^2$ | 0.1067 | 0.1341 | | | 0.2228 | | |

Note: 1) $t$ statistics in parentheses 2) *p < 0.05, **p < 0.01, ***p < 0.001

within the studied rural context, higher-income households may be more willing and able to bear the cost of public transportation, reducing their need for private vehicle ownership. Although the four study villages share similar public transport systems, higher-income individuals may utilize public transportation more frequently by prioritizing comfort, reliability, and cost-effectiveness over the long-term expenses associated with car ownership, such as fuel, insurance, maintenance, and depreciation.

Overall, the model explains about 10.67% of the variance in car ownership, suggesting that while the identified factors contribute to understanding car ownership decisions, other unexplored variables may also play a role. The negative impact of the SHRR program on car ownership highlights the importance of infrastructure improvements in shaping travel behavior and reducing the reliance on car ownership in rural areas. To effectively reduce car ownership further, complementary strategies such as enhancing public transport services or implementing travel demand management (TDM) measures could be considered, similar to the findings in other studies that emphasize the need for integrated policies to achieve substantial shifts in travel behavior.

### 4.3. Direct and indirect effects on travel frequency and distance

The Table 4 also presents the direct, indirect, and total effects of various variables on travel frequency and distance. *SHRR* has a significant direct negative effect on travel frequency, with a coefficient of -2.8012 (p < 0.01). This suggests that in areas where the SHRR program was implemented, residents tend to go out less frequently. The indirect effect of SHRR through other mediating variables is positive (0.2021), but not significant, leading to a significant total negative effect of -2.5991 (p < 0.01). This indicates that while the SHRR program might enhance certain aspects of infrastructure, it potentially reduces the necessity or frequency of travel, possibly by improving access to services locally or by reducing the time needed to reach destinations.

*Car_ownership* has a significant direct effect on average travel distance, with a coefficient of 0.5492 (p<0.01). This indicates that households with car ownership tend to have longer travel distances, reflecting the increased mobility that car ownership provides. This finding is consistent across various studies, where car ownership is often associated with higher travel distances due to the convenience and flexibility that cars offer. *SHRR* also shows a significant direct positive effect on travel distance, with a coefficient of 0.9519 (p<0.001), and a significant total effect of 0.8297 (p<0.01). This suggests that the implementation of SHRR in rural areas leads to longer travel distances. This could be because improved road infrastructure makes longer trips more feasible and efficient, encouraging residents to travel further distances. The indirect effect of SHRR through other mediating variables is negative (-0.1222), although not statistically significant. This small negative indirect effect slightly reduces the overall positive impact of SHRR on travel distance.

The results highlight the dual impact of the SHRR program on travel behavior. While the program reduces travel frequency, likely due to improved access and efficiency, it simultaneously encourages longer travel distances, perhaps because of the enhanced road infrastructure that makes longer trips more accessible. These findings underscore the importance of considering both the direct and indirect effects of infrastructure projects on travel behavior. While improving infrastructure can reduce the frequency of travel, it can also extend the reach of travel, influencing how people interact with their environment and access services in rural areas.

### 4.4. Direct and indirect effects on mode choice

The integrated Structural Equation Model (SEM) and Discrete Choice Model (DCM) analysis, as presented in Table 5, provides nuanced insights into the direct, indirect, and total effects of various variables on travel mode choice, particularly focusing on the adoption of electric bicycles and motorized modes such as cars and motorcycles.

Electric bicycles play a crucial role in rural mobility by offering a cost-effective, flexible, and accessible transportation option. Their affordability and low operational costs make them an attractive alternative to private cars and motorcycles, particularly amid rising fuel prices and maintenance expenses [44,45]. Electric bicycles provide greater convenience than public transportation, allowing users to travel on their own schedules without reliance on infrequent rural bus services. Additionally, their ease of use—requiring no formal licensing in many areas—enhances accessibility for diverse demographic groups, including women, the elderly, and younger individuals [46]. These factors collectively highlight the growing significance of electric bicycles in rural transportation systems and the necessity of integrating infrastructure development with policies that support sustainable and affordable mobility solutions.

Motorized modes, including private cars and motorcycles, remain an essential component of rural travel behavior. Private cars provide higher levels of comfort, safety, and long-distance mobility, making them the preferred option for wealthier households [47]. However, high costs related to fuel, maintenance, and insurance, coupled with improvements in public transportation and alternative mobility options, may deter car ownership, particularly among lower-income households. Meanwhile, motorcycles continue to serve as a dominant transport mode in rural areas due to their affordability, speed, and ability to navigate unpaved roads. Despite their convenience, motorcycles pose greater safety risks and contribute to higher accident rates compared to other travel modes [48]. Additionally, government regulations in some regions restricting motorcycle use for safety and environmental reasons have accelerated the shift toward electric bicycles and other alternative transport modes [45].

Car Ownership emerges as a significant determinant of travel mode choice. For electric bicycles, car ownership shows a positive direct effect (coefficient=1.0851, p<0.05), indicating that households owning cars are more likely to also use electric bicycles. However, this direct positive influence is offset by a significant negative indirect effect (-0.4954, p<0.05), leading to a non-significant total effect. This suggests a complex dynamic where car ownership facilitates electric bicycle use, but other mediating factors diminish this effect. Conversely, for motorized modes, car ownership has a substantial direct effect (2.0102, p<0.001) and an even larger total effect (3.6850, p<0.001), which is expected given that ownership of cars inherently promotes the use of motorized transport.

**Table 5. Integrated SEM and DCM model estimation for standardized direct, indirect and total effects on travel mode choice.**

| Variables | Electric bicycle | | | Motorized mode | | |
|---|---|---|---|---|---|---|
| | Direct | Indirect | Total | Direct | Indirect | Total |
| | effect | effect | effect | effect | effect | effect |
| Car_ownership | 1.0851* | −0.4954* | 0.5897 | 2.0102*** | 1.6748** | 3.6850*** |
| | (−1.66) | (−1.96) | (−0.84) | (−3.76) | (−3.27) | (−4.97) |
| Age | 0.05 | −0.0235 | 0.0265 | −0.0273 | −0.0309* | −0.0582 |
| | (−1.12) | (−1.47) | (−0.56) | (−0.83) | (−1.82) | (−1.56) |
| Education | −0.3782 | 0.1099 | −0.2683 | −0.2761 | 0.04 | −0.2361 |
| | (−1.17) | (−0.96) | (−0.78) | (−1.11) | (−0.32) | (−0.95) |
| Self_health | −0.8359 | −0.1084 | −0.9443* | −0.5008 | −0.2214 | −0.7222 |
| | (−1.66) | (−0.63) | (−1.78) | (−1.31) | (−1.17) | (−1.52) |
| Agri_work_share | −2.9136* | −0.1506 | −3.0642* | −2.0752 | −0.1473 | −2.2225 |
| | (−1.95) | (−0.35) | (−1.97) | (−1.79) | (−0.29) | (−1.49) |
| Incomelevel_22 | −1.3714** | −0.1240 | −1.4954** | −0.7088* | −0.2359 | −0.9447 |
| | (−2.81) | (−1.04) | (−2.98) | (−2.22) | (−1.85) | (−1.84) |
| BK | −0.7531 | −0.1472 | −0.9003 | 0.108 | −0.1591 | −0.0511 |
| | (−1.41) | (−1.10) | (−1.63) | (−0.38) | (−1.06) | (−0.32) |
| SHRR | 2.2859* | −1.1881** | 1.0978 | 2.2091** | −1.0785* | 1.1306 |
| | (−2.43) | (−2.41) | (−1.03) | (−2.82) | (−2.24) | (−1.01) |
| Average_travel_distance | −0.6853* | | −0.6853* | −0.4141 | | −0.4141 |
| | (−2.31) | | (−2.31) | (−1.61) | | (−1.61) |
| Average_week_times_goout | 0.1175 | | 0.1175 | 0.1074* | | 0.1074* |
| | (−1.6) | | (−1.6) | (−1.66) | | (−1.66) |
| LL$_K$/LRI | −758.76158/0.8276 | | | | | |
| AIC/Adjust BIC | 1617.523/1758.939 | | | | | |

Note: 1) *t* statistics in parentheses 2) *p<0.05, **p<0.01, ***p<0.001

SHRR Program also plays a crucial role in influencing travel mode choice. The SHRR program shows a significant positive direct effect on the use of electric bicycles (2.2859, p<0.05), suggesting that improved road infrastructure encourages the adoption of these vehicles. However, the SHRR program also exhibits a significant negative indirect effect (-1.1881, p<0.01), resulting in a non-significant total effect, indicating that while infrastructure improvements make electric bicycles more appealing, there are counteracting factors reducing their usage. For motorized modes, SHRR similarly demonstrates a significant positive direct effect (2.2091, p<0.01), with the total effect remaining slightly positive but non-significant (1.1306). This finding highlights the role of road improvements in facilitating the use of motorized vehicles, albeit with some mitigating influences.

Travel Distance and Frequency offer additional insights into travel mode preferences. The analysis shows that longer travel distances are significantly associated with reduced use of electric bicycles (-0.6853, p<0.05), likely because these vehicles are less suited for longer journeys. In contrast, the effect of travel distance on motorized mode use, while negative, is not significant, indicating that distance alone does not strongly deter the use of motorized transport. Travel frequency has a positive and significant direct effect on the use of motorized modes (coefficient=0.1074, p<0.05). This finding suggests that individuals who travel more frequently during the week are more likely to rely on motorized modes, such as cars or motorcycles. This relationship is likely driven by the need for efficiency and convenience in managing frequent trips, where motorized modes offer time savings and flexibility that are less available with non-motorized or less powerful modes like electric bicycles.

In terms of the travelers' socio-demographic factors, income level and agricultural work share are also critical factors. Higher income levels are associated with a significant decrease in the use of electric bicycles (coefficient = -1.3714, p < 0.01) and a weaker negative effect on motorized modes. This trend suggests that as income increases, households may prefer more conventional motorized vehicles over electric bicycles, possibly due to the increased affordability of cars. Similarly, reliance on agriculture (as indicated by *Agri_work_share*) shows a significant negative effect on the use of electric bicycles, reflecting that those heavily dependent on agriculture may find electric bicycles less practical or necessary.

The findings highlight the complex interplay of factors influencing travel mode choice in rural areas. Infrastructure improvements, such as those introduced through the SHRR program, play a significant role in shaping travel behavior, promoting the use of both electric bicycles and motorized modes. However, these effects are influenced by a range of socio-economic factors, including car ownership, income levels, and reliance on agriculture, which mediate and sometimes counteract the impact of infrastructure improvements.

Car ownership and the SHRR program both show significant positive effects on travel mode choice, suggesting that while these factors enhance overall mobility, they also introduce complexities in the adoption of different transportation modes. Car ownership is strongly associated with an increased likelihood of using motorized modes such as private cars and motorcycles. However, its effect on electric bicycle use is more nuanced, as improved infrastructure and greater mobility options may either encourage or diminish electric bicycle adoption, depending on factors such as travel distance, cost considerations, and individual preferences. Additionally, the significant impact of weekly travel frequency underscores the importance of understanding how daily mobility needs influence transportation choices. Frequent travelers are likely to prioritize modes of transport that can reduce travel time and provide reliable access to various destinations, particularly in rural areas where distances between locations can be substantial.

## 4.5. Robustness check

The primary analysis identified a significant relationship between the SHRR program and travel behavior. However, the interpretation of these results could be influenced by unobserved heterogeneity across villages. Since the study area consists of four distinct villages, each with unique socio-economic and infrastructural characteristics, there is a possibility that village-specific factors—such as local economic conditions, alternative transportation availability, land use patterns, or cultural attitudes toward mobility—may affect travel behavior independently of SHRR implementation.

To address these concerns, we re-estimated our models by introducing village fixed effects, using a dummy variable approach where three villages were included as categorical controls, with one village serving as the reference category. The results, presented in Tables 6 and 7, confirm the robustness of our initial findings while highlighting some variations in the magnitude of estimated effects.

Table 6 presents the standardized direct, indirect, and total effects of key variables on car ownership, travel frequency, and travel distance while controlling for village fixed effects. The results remain consistent with our initial estimation, reinforcing the negative impact of SHRR on car ownership (direct effect = -0.37, p < 0.05). The coefficient for SHRR on travel frequency remains strongly negative (direct effect = -4.638, p < 0.01; total effect = -4.319, p < 0.01) and exhibits a significant positive effect on travel distance (direct effect = 1.202, p < 0.001; total effect = 1.011, p < 0.01). The inclusion of village fixed effects does not alter our finding.

Table 7 presents the results from the integrated SEM and DCM estimation of travel mode choice with village fixed effects. The key relationships observed in the initial model remain robust, confirming that SHRR exerts a significant positive effect on electric bicycle adoption (direct effect = 4.298, p < 0.01; total effect = 3.616, p < 0.01). For motorized modes, SHRR also demonstrates a strong direct effect (3.252, p < 0.01) and total effect (2.567, p < 0.05), reinforcing the argument that enhanced rural roads encourage the use of electric bicycle and motorized mode. The shift in statistical significance for the total effect of SHRR on motorized mode adoption highlights the necessity of accounting for village-specific

**Table 6. Standardized direct, indirect and total effects on car ownership, travel frequency and travel distance with village fixed effect.**

| Variables | Car_ownership | Average_week_times_goout | | | Average_travel_distance | | |
|---|---|---|---|---|---|---|---|
| | Direct effect | Direct effect | Indirect effect | Total effect | Direct effect | Indirect effect | Total effect |
| Car_ownership | | −0.863 | | −0.863 | 0.516** | | 0.516** |
| | | (−1.19) | | (−1.19) | (2.90) | | (2.90) |
| Age | −0.013* | −0.015 | 0.011 | 0.004 | 0.009 | −0.007 | −0.002 |
| | (−2.05) | (−0.29) | (1.03) | (−0.08) | (0.72) | (−1.67) | (0.20) |
| Education | −0.045 | 0.44 | 0.039 | 0.479 | −0.088 | −0.023 | −0.111 |
| | (−0.92) | (1.11) | (0.73) | (1.21) | (−0.90) | (−0.88) | (−1.11) |
| Self_health | −0.121 | 0.41 | 0.105 | 0.514 | −0.021 | −0.062 | −0.084 |
| | (−1.62) | (0.67) | (0.96) | (0.85) | (−0.14) | (−1.41) | (−0.55) |
| Agri_work_share | −0.086 | 0.858 | 0.074 | 0.933 | −0.108 | −0.044 | −0.153 |
| | (−0.41) | (0.50) | (0.38 | (0.54) | (−0.26) | (−0.40) | (−0.35) |
| Incomelevel_22 | −0.12* | 0.207 | 0.104 | 0.311 | −0.007 | −0.062 | −0.069 |
| | (−2.40) | (0.50) | (1.07) | (0.77) | (−0.07) | (−1.85) | (−0.67) |
| BK | −0.032 | −0.071 | 0.027 | −0.044 | 0.098 | −0.016 | 0.082 |
| | (−0.52) | (−0.15) | (0.48) | (−0.09) | (0.82) | (−0.51) | (0.66) |
| SHRR | −0.37* | −4.638** | 0.3196 | −4.319** | 1.202*** | −0.191 | 1.011** |
| | (−2.09) | (−3.18) | (1.04) | (−2.99) | (3.35) | (−1.69) | (2.77) |
| Village fixed effect | YES | YES | YES | YES | YES | YES | |
| $R^2$ | 0.127 | 0.201 | | | 0.291 | | |

Note: 1) $t$ statistics in parentheses2) *p<0.05, **p<0.01, ***p<0.001

characteristics when analyzing rural mobility patterns. In summary, the robustness check with village fixed effects confirms that SHRR significantly influences travel mode choices, promoting both electric bicycle and motorized mode adoption.

## 5. Conclusion

This study set out to examine the impact of the SHRR Program on travel behavior in rural areas, specifically focusing on how infrastructure improvements influence travel mode choice, travel frequency, and travel distance. Using an integrated SEM and DCM framework, the research analyzed data from 127 households across four villages in Qingshuihe County, Inner Mongolia, China collected between August 2023 and March 2024. The study incorporated key socio-demographic variables, such as car ownership, income levels, education, and self-reported health, to provide a comprehensive understanding of the factors that shape mobility patterns in these rural settings.

The findings indicate that the SHRR program has a multifaceted impact on travel behavior. Specifically, the program significantly reduces travel frequency, a result likely attributable to improved local accessibility, which decreases the necessity for frequent trips. This suggests that as infrastructure enhances, residents may be able to accomplish more within a single trip, reducing the need for multiple journeys. Concurrently, the SHRR program facilitates longer travel distances and increases reliance on both motorized modes and electric bicycles, reflecting the broader mobility shifts enabled by rural road enhancements. The improved road infrastructure not only makes longer trips more feasible but also encourages the adoption of private cars, motorcycles, and electric bicycles, as these modes offer greater flexibility and efficiency for covering extended distances. Car ownership emerges as a pivotal factor in this context, showing strong positive effects on both travel distance and the likelihood of choosing motorized modes. Households with access to these transport options are better equipped to take advantage of the improved infrastructure, resulting in more frequent and longer trips. However, the relationship between car ownership and the use of electric bicycles is

**Table 7. Integrated SEM and DCM model estimation for standardized direct, indirect and total effects on travel mode choice with village fixed effect.**

| Variables | Electric bicycle | | | Motorized mode | | |
|---|---|---|---|---|---|---|
| | Direct | Indirect | Total | Direct | Indirect | Total |
| | effect | Effect | effect | effect | effect | effect |
| Car_ownership | 1.46 | −0.231 | 1.229 | 2.235*** | −0.131 | 2.103*** |
| | (1.84) | (−0.94) | (1.59) | (3.40) | (−0.63) | (3.35) |
| SHRR | 4.298** | −0.682 | 3.616** | 3.252** | −0.685 | 2.567* |
| | (3.05) | (−1.21) | (2.72) | (2.80) | (−1.27) | (2.31) |
| Average_travel_distance | −0.41 | | −0.41 | −0.19 | | −0.19 |
| | (−1.16) | | (−1.16) | (−0.61) | | (−0.61) |
| Average_week_times_goout | 0.006 | | 0.006 | 0.029 | | 0.029 |
| | (0.07) | | (0.07) | (0.38) | | (0.38) |
| Household characteristics | Yes | Yes | Yes | Yes | Yes | Yes |
| Village fixed effect | Yes | Yes | Yes | Yes | Yes | Yes |
| LL$_K$/LRI | −745.5922/0.8245 | | | | | |
| AIC/Adjust BIC | 267.687/276.22 | | | | | |

Note: 1) *t* statistics in parentheses2) *p<0.05, **p<0.01, ***p<0.001

more complex. While infrastructure and car ownership collectively enhance mobility, they may simultaneously reduce the necessity or appeal of certain transport modes, particularly electric bicycles, depending on trip purpose, cost considerations, and individual travel preferences. These findings highlight the dual impact of infrastructure improvements: while fostering greater mobility and travel efficiency, they also reshape transport mode preferences. This study contributes to the understanding of rural mobility dynamics by providing empirical evidence on how infrastructure interventions influence travel behaviors, emphasizing the need for integrated policy approaches that account for both direct and indirect effects on transport choices.

Despite the valuable insights provided by this study, several limitations should be acknowledged. First, the sample size is relatively small (127 households across four villages), which may limit the generalizability of the findings to other rural areas in China or beyond. While the inclusion of village fixed effects improves robustness, additional research with a larger and more diverse sample could help validate and refine the observed relationships. Second, although the study employs an integrated SEM and DCM framework to examine the direct and indirect effects of SHRR on travel behavior, potential endogeneity issues remain. Factors such as unobserved individual preferences, self-selection bias, and unmeasured policy interventions could influence transport choices. Future studies could benefit from instrumental variable approaches or longitudinal data to better isolate causal relationships. Third, while this study highlights the role of electric bicycles, private cars, and motorcycles, it does not delve into detailed household decision-making processes regarding transport mode adoption. Future research could explore qualitative factors, such as perceptions of convenience, safety, affordability, and environmental sustainability, to complement quantitative modeling.

## Supporting information

**S1 File. Original ethics approval letter.**
(PDF)

**S2 File. English version of the original ethics approval letter.**
(PDF)

**S3 File. PLOS' questionnaire on inclusivity in global research.**
(DOCX)

**S4 File. Research data 2.**
(ZIP)

## Author contributions

**Conceptualization:** Yintu Bao, Ligao Bao.

**Data curation:** Yintu Bao.

**Formal analysis:** Yintu Bao, Ligao Bao.

**Funding acquisition:** Yintu Bao.

**Methodology:** Yintu Bao, Ligao Bao.

**Resources:** Yintu Bao.

**Supervision:** Ligao Bao.

**Validation:** Gumula Wuri, Dan Shan.

**Writing – original draft:** Ligao Bao.

**Writing – review & editing:** Gumula Wuri, Dan Shan, Ligao Bao.

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
