## [Decision Letter · Decision Letter 0]

6 Jan 2025

PONE-D-24-41259Paving the Way for Rural Revitalization: Empirical Analysis of the ‘Sihao Rural Road Policy’ and Transport Mode Choice in Inner Mongolia, ChinaPLOS ONE

Dear Dr. Bao,

Thank you for submitting your manuscript to PLOS ONE. After careful consideration, we feel that it has merit but does not fully meet PLOS ONE’s publication criteria as it currently stands. Therefore, we invite you to submit a revised version of the manuscript that addresses the points raised during the review process.

Reviewer 1 recommended minor revision while Reviewer 2 recommended reject. Both reviewers pointed out poor descriptions about the study efforts and achievements. I hope the authors will upgrade their manuscript by addressing the comments from them. If the authors would fail to significantly improve the paper, the paper should be rejected in the second-round review.

We look forward to receiving your revised manuscript.

Kind regards,

Hironori Kato, Dr. Eng.

Academic Editor

PLOS ONE

Journal Requirements:

This work was supported by the Inner Mongolia Open University Special Project, "Research on the Impact of 'Sihao Rural Roads' Construction on Rural Economy in Inner Mongolia" [grant number IMOU-SSR2302].

5. We note that you have indicated that there are restrictions to data sharing for this study. PLOS only allows data to be available upon request if there are legal or ethical restrictions on sharing data publicly. For more information on unacceptable data access restrictions, please see http://journals.plos.org/plosone/s/data-availability#loc-unacceptable-data-access-restrictions. 

6. In the online submission form, you indicated that data will be made available on request.

7. We note that Figure 2 in your submission contain [map/satellite] images which may be copyrighted. All PLOS content is published under the Creative Commons Attribution License (CC BY 4.0), which means that the manuscript, images, and Supporting Information files will be freely available online, and any third party is permitted to access, download, copy, distribute, and use these materials in any way, even commercially, with proper attribution. For these reasons, we cannot publish previously copyrighted maps or satellite images created using proprietary data, such as Google software (Google Maps, Street View, and Earth). For more information, see our copyright guidelines: http://journals.plos.org/plosone/s/licenses-and-copyright.

8. Please remove all personal information, ensure that the data shared are in accordance with participant consent, and re-upload a fully anonymized data set. 

Reviewers' comments:

Reviewer's Responses to Questions

**Comments to the Author**

1. Is the manuscript technically sound, and do the data support the conclusions?

Reviewer #1: Partly

Reviewer #2: Partly

2. Has the statistical analysis been performed appropriately and rigorously? 

Reviewer #1: No

Reviewer #2: No

3. Have the authors made all data underlying the findings in their manuscript fully available?

Reviewer #1: Yes

Reviewer #2: No

4. Is the manuscript presented in an intelligible fashion and written in standard English?

Reviewer #1: Yes

Reviewer #2: Yes

5. Review Comments to the Author

Reviewer #1: - Thank you very much for giving us the opportunity to review this interesting paper. The paper’s value is to assess the effects of a rural road program on people’s travel behaviors after rural road improvement. The following could be addressed to strengthen the discussions in the paper and help readers’ understanding.

- Please ensure that the paper should be proofread carefully during revision for second review.

- 2.1. Please clarify why this paper needs to focus on travel behaviors when you measure the impact of “Sihao Rural Road.” This explanation is needed to justify your 3 key questions in Section 4.1.

- Please include the potential effects of the relatively small sample size (127 households) on the overall modeling results, probably in Section 5. Furthermore, please provide challenges you faced during the data collection, which resulted in the removal of 73 sample households.

- 3.3. Please describe briefly each village's socio-economic characteristics and road density. In the modeling exercise, village-level variables were not included. Villages might be influential factors to people's livelihoods and travel behavioral patterns, however, the study villages' differences are not detailed other than whether the Sihao Rural Road program has been applied or not.

- 4.1.

a. Please add the discussion on whether any other government programs or policies or any infrastructure development projects were implemented in parallel with the Sihao Rural Road program. If any of the programs/policies/projects existed, please explain how you eliminated those programs/policies’ effects on the variables for this paper, particularly income level, agricultural sectors, health conditions, education, and travel modal ownership.

b. The information in Table 2 does not support the descriptive explanation of the sample in Section 4.1. Please check the internal linkages. Major items are pointed out below for your consideration.

c. Please add the discussion on the average and mean income and the economic dependency on the agriculture sector of the study country and each study village, as these factors are influential to people’s decision to obtain transport means, particularly motorized ones. Also, the discussion will substantiate “the sample is primarily composed of low- to middle-income households” after Table 2.

d. Please provide statical information in the table to support “Education data reveal that 37.01% of respondents….primary school education.”

e. “Average” and “mean” are mixed among the table and main texts. Please ensure consistency.

f. Please describe the average market prices of major transport means in the country and, if data is available, each village.

g. How did you consider villages' influences? You may include a dummy variable representing each village at modelling exercises. Furthermore, the descriptive statistics in Table 2 can be divided for each village, which should be possible as the total sample number is 127.

- 4.2. It is interesting that the rural road policy would negatively impact people’s car ownership. However, the interpretation of modeling results should be carefully made. Please include the discussion of potential external factors, such as prices or market conditions of cars, development of public transportation or shared transportation services, and so on. The discussion is needed to support your finding (within the studied rural context, higher-income households might have better access to alternative modes of transportation, or they might reside in areas where car ownership is less necessary due to better connectivity or availability of services) and need for complementary strategies as described in the last paragraph in this section. Also, I observe a potential contradiction with the implication in Section 4.3 (Car ownership further amplifies this effect by providing the means to travel further). Please check internal consistency.

- 4.4. Will you add the roles of electric bicycles in people’s lives in the country and people’s motivations to buy them? Electric bicycles are focused too much in this section, but you may have some ideas to justify this. One reason for the potential unclarity of this section is the absence of each travel mode’s role in people’s lives in the country, and another reason is that “motorized mode” is not clearly defined in the paper’s contexts (sometimes. I read that “motorized mode” means only car, but not always). Please give an in-depth analysis of each travel mode. This ambiguity makes the last paragraph difficult to read and the conclusion statement in Section 5.

Reviewer #2: In comparing the studies in the four villages, it is undeniable that the differences in the transportation behavior of the people in these villages are due to conditions other than the presence or absence of the SHRR Policy.The SEM and the Descrete choice model framework and its thin description of the results do not allow us to judge the validity of this model and results. Overall, the simplicity of the survey makes it difficult to conclude the impact of the SHRR Policy from this alone.

6. PLOS authors have the option to publish the peer review history of their article (what does this mean? ). If published, this will include your full peer review and any attached files.

**Do you want your identity to be public for this peer review?** For information about this choice, including consent withdrawal, please see our Privacy Policy .

Reviewer #1: No

Reviewer #2: No

---

## [Author Response · Author response to Decision Letter 1]

3 Mar 2025

Reviewers' comments:

Reviewer's Responses to Questions

Comments to the Author

1. Is the manuscript technically sound, and do the data support the conclusions?The manuscript must describe a technically sound piece of scientific research with data that supports the conclusions. Experiments must have been conducted rigorously, with appropriate controls, replication, and sample sizes. The conclusions must be drawn appropriately based on the data presented.

Reviewer #1: Partly

Reviewer #2: Partly

Response: We acknowledge the reviewers’ concerns and have carefully revised the manuscript to ensure that the analysis aligns with the conclusions. We have refined the discussion, improved the interpretation of findings, and added robustness checks, including village fixed effects, to strengthen the validity of the results.

2. Has the statistical analysis been performed appropriately and rigorously?

Reviewer #1: No

Reviewer #2: No

Response: We have revised our statistical approach to enhance its rigor. Specifically, we have:

• Included village fixed effects to account for potential unobserved heterogeneity.

• Clarified the model specification and addressed any inconsistencies in the interpretation of direct and indirect effects.

• Improved the presentation of statistical results, ensuring clarity and transparency in model estimations.

3. Have the authors made all data underlying the findings in their manuscript fully available?

Reviewer #1: Yes

Reviewer #2: No

Response: We acknowledge Reviewer #2’s concern regarding data availability. The dataset used in this study contains information that cannot be publicly shared due to privacy and confidentiality constraints. However, the data will be made available upon reader’s reasonable request to the corresponding author, in line with ethical and institutional guidelines.

5. Review Comments to the Author

Reviewer #1: - Thank you very much for giving us the opportunity to review this interesting paper. The paper’s value is to assess the effects of a rural road program on people’s travel behaviors after rural road improvement. The following could be addressed to strengthen the discussions in the paper and help readers’ understanding.

Response: Thank you for your encouraging comments and for recognizing the contribution of our manuscript. We appreciate your careful reading. Below, we address your specific suggestion in detail:

- Please ensure that the paper should be proofread carefully during revision for second review.

Response: We have carefully proofread the manuscript and made several revisions to improve its clarity, readability, and academic tone. Specific grammatical, structural, and typographical errors have been corrected to enhance the overall quality of the paper.

- 2.1. Please clarify why this paper needs to focus on travel behaviors when you measure the impact of “Sihao Rural Road.” This explanation is needed to justify your 3 key questions in Section 4.1.

Response: To improve the logical flow and strengthen the justification for analyzing travel behavior, we have moved the three key research questions from Section 4.1 (Line:401-413) to the Introduction section. (Line:121-138)

We utilize an integrated Structural Equation Modeling (SEM) and Discrete Choice Modeling (DCM) framework to capture both the direct and indirect effects of the SRRP on travel behavior, thus offering a more nuanced understanding of the mechanisms behind these changes.

In particular, the research addresses the following questions:

(1) Does the SRRP continue to exert significant direct effects on car ownership, travel frequency, travel distance, and travel mode choice after controlling for socio-demographic factors?

(2) How does the SRRP indirectly influence travel mode choice through its effects on mediating variables such as car ownership, travel frequency, and travel distance?

(3) What is the overall impact of the SRRP on travel mode choice, combining both direct and indirect effects?

By examining these questions, the study aims to provide a comprehensive perspective on how rural infrastructure improvements like the SRRP shape travel behavior. Findings from this research can inform policy recommendations for future rural road development and identify ways to optimize infrastructure investments to achieve broader socio-economic benefits and sustainable mobility patterns.

Additionally, we have explicitly clarified why measuring the impact of the Sihao Rural Road (SHRR) program necessitates a focus on travel behaviors. (Line:88-115)

The decision to focus on travel behavior stems from the critical role that daily mobility plays in realizing the broader benefits of rural road improvements. Enhanced rural roads can fundamentally reshape how often residents travel, how far they go, and which modes of transport they prefer. These changes in travel behavior mediate a wide range of socioeconomic outcomes: improved roads may reduce the time and financial costs of commuting, prompt shifts in mode preference (e.g., from walking to motorized modes), and influence household decisions about vehicle ownership. By capturing these direct and indirect pathways, researchers gain a more comprehensive understanding of how the SRRP translates into real-world changes in mobility patterns—ultimately affecting economic opportunities, access to services, and social inclusion.

The SRRP, initiated in 2015, is particularly interesting because it aims to fix the differences in transport infrastructure between urban and rural areas (MOT, 2015). By focusing on improving the quality and connectivity of rural roads, the policy aspires not only to make travel more convenient but also to foster new economic opportunities in rural areas. This could help close the persistent urban–rural income gap, a central concern for sustainable development. Through enhanced transport links, rural communities could see boosts in employment, market access, and overall well-being (MOT, 2019). Despite the program’s potential, there is still limited empirical evidence demonstrating how specific aspects of the SRRP—like upgraded road infrastructure and enhanced road management—translate into shifts in travel behavior and ultimately contribute to rural revitalization.

This revision ensures a stronger connection between our research objectives and the empirical analysis.

- Please include the potential effects of the relatively small sample size (127 households) on the overall modeling results, probably in Section 5. Furthermore, please provide challenges you faced during the data collection, which resulted in the removal of 73 sample households.

Response: We appreciate the reviewer’s concern about the sample size and have added the following discussion to Section 3.3 and 5:

1. Data Collection Challenges: (Line:291-296)

We used a stratified sampling technique to select 200 households for the survey, which was distributed across four villages village A, village B, village C, and village D in Qingshuihe County. However, due to incomplete questionnaires (n = 10), data inconsistencies (n = 18), and non-responses (n = 45), a total of 73 responses were excluded. As a result, the final sample for analysis comprised 127 valid responses.

2. Sample Size Limitations:

While the final sample of 127 households provided sufficient statistical power to detect medium-to-large effect sizes within the structural equation modeling (SEM) framework, it may have limited the detection of smaller interaction effects. To acknowledge this limitation, we have added a discussion in Section 5, emphasizing the need for future studies with larger and more diverse samples to enhance the robustness of the findings. (Line:634-638)

Despite the valuable insights provided by this study, several limitations should be acknowledged. First, the sample size is relatively small (127 households across four villages), which may limit the generalizability of the findings to other rural areas in China or beyond. While the inclusion of village fixed effects improves robustness, additional research with a larger and more diverse sample could help validate and refine the observed relationships.

- 3.3. Please describe briefly each village's socio-economic characteristics and road density. In the modeling exercise, village-level variables were not included. Villages might be influential factors to people's livelihoods and travel behavioral patterns, however, the study villages' differences are not detailed other than whether the Sihao Rural Road program has been applied or not.

Response: Thank you for your suggestion, to address this concern, we have added a detailed description of the socio-economic characteristics and road density of each study village in Section 3.3 (The study area). (Line:303-321)

Table 1 presents key characteristics for the four study villages. Village A shows a moderate road density (1.15 km/km²), lower average income (14,500 yuan), and high reliance on agriculture (81%), suggesting a stronger dependence on traditional, local transport modes. Village B, which did not implement the SHRR, has a low road density (0.71 km/km²) but a high paved road percentage (93%) and an average income of 16,500 yuan—indicating that prior infrastructure investments have improved road quality, even if internal connectivity remains limited. In contrast, Village C stands out with the highest road density (2.3 km/km²), the lowest transport cost (5 yuan) and travel time (5 minutes) to the county center, an average income of 15,000 yuan, and a lower agricultural labor share (65%), reflecting superior connectivity and accessibility. Village D, which implemented the SHRR, shares a low road density (0.71 km/km²) with Village B but has the highest average income (17,000 yuan) and a moderate agricultural labor share (80%), along with higher transport costs (15 yuan) and longer travel time (20 minutes). For broader context, the average income in Qingshuihe County (24,110 yuan) is notably higher than that of the study villages, indicating economic disparities between rural and county-level populations. Moreover, the county’s overall road density (0.65 km/km²) is lower than that of all study villages, as much of the county consists of non-residential land such as mountains, forests, and agricultural fields. This suggests that while rural areas face infrastructural challenges, localized road investments, including the SHRR program, may have enhanced connectivity in specific villages.

- 4.1.

a. Please add the discussion on whether any other government programs or policies or any infrastructure development projects were implemented in parallel with the Sihao Rural Road program. If any of the programs/policies/projects existed, please explain how you eliminated those programs/policies’ effects on the variables for this paper, particularly income level, agricultural sectors, health conditions, education, and travel modal ownership.

Response: We appreciate your suggestion and acknowledge that infrastructure improvements and policy interventions do not occur in isolation. To address this concern, we have incorporated a discussion in Section 3.3 regarding other government programs and policies that were implemented alongside the SHRR program. (Line:323-335)

In the study area of Qingshuihe County, government-led initiatives such as targeted poverty alleviation, healthcare subsidies, and improvements to local schools and agricultural facilities have been implemented in recent years alongside the SHRR Program. To minimize potential confounding effects from these parallel policies, we employed two approaches. First, the sample villages were deliberately selected to include both households benefiting from SHRR improvements and those that did not, ensuring relative uniformity in exposure to other initiatives. Informal interviews with local officials and community leaders further confirmed that no major road-related or large-scale development programs were introduced during the study period that could disproportionately affect travel behavior. Second, the household survey included questions on recent policy-driven changes in income, agricultural activities, health services, education access, and vehicle ownership, enabling us to detect any unusual shifts unrelated to SHRR. These measures help ensure that our analysis more accurately isolates the impact of the SHRR program from other local influences.

Furthermore, we introduced Section 4.5: Robustness Check to validate our findings by integrating village-level fixed effects into the model estimation. This additional analysis helps isolate the impact of the SHRR program by controlling for unmeasured village-specific characteristics that may influence car ownership, travel behavior, and mode choice. (Line:565-597)

4.5. Robustness check

The primary analysis identified a significant relationship between the SHRR program and travel behavior. However, the interpretation of these results could be influenced by unobserved heterogeneity across villages. Since the study area consists of four distinct villages, each with unique socio-economic and infrastructural characteristics, there is a possibility that village-specific factors—such as local economic conditions, alternative transportation availability, land use patterns, or cultural attitudes toward mobility—may affect travel behavior independently of SHRR implementation.

To address these concerns, we re-estimated our models by introducing village fixed effects, using a dummy variable approach where three villages were included as categorical controls, with one village serving as the reference category. The results, presented in Table 6 and 7, confirm the robustness of our initial findings while highlighting some variations in the magnitude of estimated effects.

Table 6 presents the standardized direct, indirect, and total effects of key variables on car ownership, travel frequency, and travel distance while controlling for village fixed effects. The results remain consistent with our initial estimation, reinforcing the negative impact of SHRR on car ownership (direct effect = -0.37, p < 0.05). The coefficient for SHRR on travel frequency remains strongly negative (direct effect =-4.638, p < 0.01; total effect == -4.319, p < 0.01) and exhibits a significant positive effect on travel distance (direct effect =1.202, p < 0.001; total effect = 1.011, p < 0.01). The inclusion of village fixed effects does not alter our finding.

Table 7 presents the results from the integrated SEM and DCM estimation of travel mode choice with village fixed effects. The key relationships observed in the initial model remain robust, confirming that SHRR exerts a significant positive effect on electric bicycle adoption (direct effect = 4.298, p < 0.01; total effect = 3.616, p < 0.01). For motorized modes, SHRR also demonstrates a strong direct effect (3.252, p < 0.01) and total effect (2.567, p < 0.05), reinforcing the argument that enhanced rural roads encourage the use of electric bicycle and motorized mode. The shift in statistical significance for the total effect of SHRR on motorized mode adoption highlights the necessity of accounting for village-specific characteristics when analyzing rural mobility patterns. In summary, the robustness check with vi

---

## [Decision Letter · Decision Letter 1]

21 Apr 2025

Paving the Way for Rural Revitalization: Empirical Analysis of the ‘Sihao Rural Road Policy’ and Transport Mode Choice in Inner Mongolia, China

PONE-D-24-41259R1

Dear Dr. Bao,

We’re pleased to inform you that your manuscript has been judged scientifically suitable for publication and will be formally accepted for publication once it meets all outstanding technical requirements.

Kind regards,

Hironori Kato, Dr. Eng.

Academic Editor

PLOS ONE

Additional Editor Comments:

Both reviewers recommended accept. Congratulations for acceptance.

Reviewers' comments:

Reviewer's Responses to Questions

**Comments to the Author**

1. If the authors have adequately addressed your comments raised in a previous round of review and you feel that this manuscript is now acceptable for publication, you may indicate that here to bypass the “Comments to the Author” section, enter your conflict of interest statement in the “Confidential to Editor” section, and submit your "Accept" recommendation.

Reviewer #1: All comments have been addressed

Reviewer #2: All comments have been addressed

2. Is the manuscript technically sound, and do the data support the conclusions?

Reviewer #1: Yes

Reviewer #2: Partly

3. Has the statistical analysis been performed appropriately and rigorously? 

Reviewer #1: Yes

Reviewer #2: Yes

4. Have the authors made all data underlying the findings in their manuscript fully available?

Reviewer #1: Yes

Reviewer #2: No

5. Is the manuscript presented in an intelligible fashion and written in standard English?

Reviewer #1: Yes

Reviewer #2: Yes

6. Review Comments to the Author

Reviewer #1: Principally, the revised manuscript is well-written technically.

However, proofreading throughout the publication is still considered necessary prior to publication.

Reviewer #2: (No Response)

7. PLOS authors have the option to publish the peer review history of their article (what does this mean? ). If published, this will include your full peer review and any attached files.

**Do you want your identity to be public for this peer review?** For information about this choice, including consent withdrawal, please see our Privacy Policy .

Reviewer #1: No

Reviewer #2: No

---

## [Editor Report · Acceptance letter]

PONE-D-24-41259R1

PLOS ONE

Dear Dr. Bao,

I'm pleased to inform you that your manuscript has been deemed suitable for publication in PLOS ONE. Congratulations! Your manuscript is now being handed over to our production team.

Kind regards,

on behalf of

Dr. Hironori Kato

Academic Editor

PLOS ONE